# Contrasting Dynamics of Intracellular and Extracellular Antibiotic Resistance Genes in Response to Nutrient Variations in Aquatic Environments

**DOI:** 10.3390/antibiotics13090817

**Published:** 2024-08-28

**Authors:** Lele Liu, Xinyi Zou, Yuan Cheng, Huihui Li, Xueying Zhang, Qingbin Yuan

**Affiliations:** 1College of Environmental Science and Engineering, Nanjing Tech University, Nanjing 211816, China; leleliu0221@163.com (L.L.); zou.xinyi@foxmail.com (X.Z.); cy1881579@163.com (Y.C.); 18020117150@163.com (H.L.); 2State Key Laboratory of Pollution Control and Resource Reuse, School of the Environment, Nanjing University, Nanjing 210023, China

**Keywords:** nutrient, antibiotic resistance genes, extracellular antibiotic resistance genes, river, microcosm, ARGs-nutrient model

## Abstract

The propagation of antibiotic resistance in environments, particularly aquatic environments that serve as primary pathways for antibiotic resistance genes (ARGs), poses significant health risks. The impact of nutrients, as key determinants of bacterial growth and metabolism, on the propagation of ARGs, particularly extracellular ARGs (eARGs), remains poorly understood. In this study, we collected microorganisms from the Yangtze River and established a series of microcosms to investigate how variations in nutrient levels and delivery frequency affect the relative abundance of intracellular ARGs (iARGs) and eARGs in bacterial communities. Our results show that the relative abundance of 7 out of 11 representative eARGs in water exceeds that of iARGs, while 8 iARGs dominate in biofilms. Notably, iARGs and eARGs consistently exhibited opposite responses to nutrient variation. When nutrient levels increased, iARGs in the water also increased, with the polluted group (COD = 333.3 mg/L, COD:N:P = 100:3:0.6, m/m) and the eutrophic group (COD = 100 mg/L, COD:N:P = 100:25:5, m/m) showing 1.2 and 3.2 times higher levels than the normal group (COD = 100 mg/L, COD:N:P = 100:10:2, m/m), respectively. In contrast, eARGs decreased by 6.7% and 8.4% in these groups. On the other hand, in biofilms, higher nutrient levels led to an increase in eARGs by 1.5 and 1.7 times, while iARGs decreased by 17.5% and 50.1% in the polluted and eutrophic groups compared to the normal group. Moreover, while increasing the frequency of nutrient delivery (from 1 time/10 d to 20 times/10 d) generally did not favor iARGs in either water or biofilm, it selectively enhanced eARGs in both. To further understand these dynamics, we developed an ARGs-nutrient model by integrating the Lotka–Volterra and Monod equations. The results highlight the complex interplay of bacterial growth, nutrient availability, and mechanisms such as horizontal gene transfer and secretion influencing ARGs’ propagation, driving the opposite trend between these two forms of ARGs. This contrasting response between iARGs and eARGs contributes to a dynamic balance that stabilizes bacterial resistance levels amid nutrient fluctuations. This study offers helpful implications regarding the persistence of bacterial resistance in the environment.

## 1. Introduction

The rampant misuse of antibiotics has catalyzed a global increase in bacterial resistance [1,2], posing a formidable challenge to human health. This issue extends beyond the confines of clinical settings, permeating environmental spheres that serve as reservoirs for resistance, thereby facilitating the proliferation of antibiotic resistance. In China, the detection levels of various antibiotics in major watersheds are alarmingly high. For instance, macrolide antibiotics have been detected at concentrations ranging from 7.6 to 12,159.4 ng/L, and tetracyclines from 14.1 to 1676.9 ng/L [3,4,5]. Meanwhile, ARGs have also been detected across diverse environmental media [3,6], with a significant presence particularly in river ecosystems [6,7,8], manifesting in both planktonic and biofilm forms, with the relative abundances being up to 10^−2^ copies/16srDNA [9,10]. ARGs in river ecosystems can be transmitted to humans through various routes, such as drinking water, agricultural irrigation, and animal husbandry, posing great risks to human health. However, research on the key influences on the reproduction and transfer of ARGs in actual water environments is scanty, and there is an urgent need for a comprehensive understanding of the fate of ARGs and the means of limiting their transmission in river systems.

The emergence and propagation of ARGs in aquatic environments are influenced by a complex interaction of biotic and abiotic factors [11], with nutrients playing a pivotal role. Nutrients are vital for the survival and replication of hosts harboring ARGs, as the expression and replication of ARGs are energy-dependent processes. Consequently, the availability of nutrients is crucial for these hosts to offset adaptation costs [12]. For example, it has been reported that the adaptation cost of host-carrying ARGs increases as nutrient availability decreases [13]. This is because hosts carrying ARGs and their non-resistant counterparts often compete for the same nutrients, leading to heightened interspecific competition as nutrient availability declines [14,15]. Not only does this influence ARG-carrying hosts, but nutrient levels also modulate the sensitivity of antibiotic-sensitive bacteria to antibiotic exposure. It has been reported that, when antibiotics are dosed into the system at a certain frequency, synchronization of this frequency with nutrient dosing promotes the proliferation of antibiotic-resistant bacteria, whereas an asynchrony between the two favors the development of sensitive bacteria [13]. Furthermore, the fluctuation of nutrient levels, rather than their mere abundance, plays a significant role in shaping antibiotic resistance patterns by increasing the adaptive cost of antibiotic-resistant bacteria and decreasing their abundance. This leads to disadvantage during competition [13]. However, these conclusions are predominantly based on pure cultured laboratory conditions or theoretical modeling analysis. The impact of nutrient concentrations and exposure frequencies on natural aquatic microflora remains poorly understood, highlighting the need for microcosm studies or field research that accurately replicates actual environmental conditions.

Thus, we hypothesize intracellular ARGs (iARGs) within bacterial hosts but also extracellular ARGs (eARGs) that exist outside the cells. iARGs are housed within cells but, when cells undergo lysis or secretion, ARGs are released into the environment, where they can persist for extended periods, especially when adsorbed onto particles [16], with their longevity ranging from weeks to years. These eARGs can be assimilated by non-resistant bacteria through natural transformation, representing an additional pathway for the spread of resistance [17], and are an important part of the arsenal of ARGs in the environment. In various environments, such as wastewater effluents, aerosols, and rivers, the abundance of eARGs is comparable to or even exceeds that of iARGs [12,18,19], highlighting the significance of the switch between iARGs and eARGs in the transmission and risk of ARGs. Given that nutrients can affect the survival, activity, and propagation of bacteria hosting ARGs, the potential of these bacteria to produce eARGs, by secretion or cell lysis, known as the two dominant pathways of eDNA production [20], should also be affected by nutrients.

Thus, we hypothesize that nutrients will influence the development of bacterial resistance, including both iARGs and eARGs, in distinct ways due to the differing characteristics of these two types of ARGs. To test this hypothesis, we developed a series of microcosms using microflora collected from the Yangtze River, the largest river in Asia. We then explored the colonization and development of intracellular and extracellular ARGs in planktonic and biofilm forms under varying levels of organic matter, nitrogen, and phosphorus nutrients, including conditions representative of normal river water (COD = 100 mg/L, COD:N:P = 100:10:2, m/m), polluted water (COD = 333.3 mg/L, COD:N:P = 100:3:0.6, m/m), and eutrophic water (COD = 100 mg/L, COD:N:P = 100:25:5, m/m)*,* reflecting the water quality found in rivers under different conditions [21,22]. Additionally, to account for the dynamic nature of nutrient fluctuations in real environments, such as the intermittent discharge of wastewater into rivers, we simulate nutrient dosing frequencies, including single administration (1 time/10 d) and split administration (5 times/10 d and 20 times/10 d), while maintaining consistent total nutrient content. To unravel the underlying mechanisms, an innovative mathematic ARGs-nutrient model describing the effects of nutrients on ARGs propagation was developed based on the Lotka–Volterra model and Monod equation [23,24]. The translational relationship between the development of ARGs and nutrient conditions provides valuable insights into the persistence of bacterial resistance in real-world environments. These findings also offer important implications for reducing the potential risk of ARG spread to humans and ecosystems.

## 2. Results and Discussion

### 2.1. Dominance of eARGs in Water and iARGs in Biofilms 

To gain insights into the ARG profile, the relative abundance of iARGs and eARGs in the Yangtze River prior to microcosm experimentation was investigated (Figure 1 and Appendix A). The results reveal that eARGs are more prevalent in water compared to iARGs, with seven out of eleven ARGs being present in this form. Notably, eARGs associated with *bla_TEM_* and *tetG* genes account for over 90% of the total eARG abundance. This observation indicates that eARGs are widespread and prevalent in aquatic environments and their relative abundance is higher than that of iARGs, which is consistent with the results of previously conducted studies [25,26]. Previous research has identified eARGs in various aquatic settings, including rivers [19], marine sediments [25], sewage treatment plants [27,28], and medical wastewater [29]. eARGs are likely attributed to the lysates of dead cells and secretions from living cells [30,31]. Moreover, eDNA can bind to soil colloids, sand, clay minerals, and humic substances, thereby protecting against nuclease degradation [32]. These active eARGs can be acquired by recipient cells through horizontal gene transfer (transformation) [19].

In contrast, iARGs are the most prevalent form of ARGs in biofilms, with eight out of eleven ARGs present in this form. Biofilms, typically rich in nutrients [33,34], provide an environment conducive to the persistence and rapid multiplication of resistant bacteria with reduced fitness costs. On the other hand, extracellular ARGs (eARGs) are less likely to be released from biofilms [35]. Furthermore, even when eARGs are released, they are more likely to dissolve in water rather than adhere to biofilms.

### 2.2. Nutrient Variations Induce Opposite Responses in iARGs and eARGs

While several studies have focused on the impact of nutrients on iARGs, the response of extracellular eARGs and the transition between iARGs and eARGs remains largely unexplored. In this study, we established three scenarios: a normal group representing typical water conditions in rivers, a polluted group reflecting conditions frequently reported in urban inland rivers, and a eutrophic group characterizing nutrient-rich environments (Figure 2) [36,37]. Compared to the normal group, both the polluted and eutrophic groups exhibited higher abundances of iARGs in water, with increases of 1.2 and 3.2 times (*p* < 0.05), respectively. This suggests that elevated nutrient levels favor the propagation of bacteria hosting ARGs within cells. However, the relative abundance of eARGs in water slightly decreased in the polluted and eutrophic groups, reaching 93.3% and 91.6% of the normal group’s levels (*p* > 0.05), respectively. This indicates a distinct pattern of eARG production, despite all eARGs originating from the lysis or secretion of bacteria harboring ARGs [38].

In biofilms, increased nutrient levels did not result in a rise in iARG abundance. On the contrary, the relative abundance of iARGs in biofilms generally decreased with increasing nutrient concentration (*p* < 0.05). Specifically, the mean value of the eleven iARGs’ abundance for the polluted group decreased by 17.5% compared to the normal group, while the mean value of the eleven iARGs for the eutrophic group decreased by 50.1% (*p* < 0.05). For eARGs, the trend was opposite to that observed in water. The relative abundance of eARGs in the polluted and eutrophic groups increased by 1.5–1.7 times compared to the normal group (*p* < 0.05). These findings highlight the complex variation in ARGs’ abundance in real riverine systems, which are linked to factors such as the spreading of ARGs, bacterial cleavage and secretion of ARGs, transformation of ARGs, and dynamic conversion of ARGs in the water and biofilms. To further investigate the underlying mechanisms, a comprehensive model was developed.

To explore the mechanism of nutrient levels affecting iARGs and eARGs, we developed an ARGs-nutrient model to study ARG propagation under varying conditions of nutrient abundance (Figure 3). The results showed that the relative abundance of iARGs in water tended to increase with elevated carbon (from 100 mg/L to 500 mg/L) and nitrogen levels (from 5 mg/L to 25 mg/L) (*p* > 0.05). This trend, aligning with the microcosm experiment, is likely to be due to nutrient promoting the proliferation of drug-resistant bacteria while reducing their adaptive fitness costs, a phenomenon consistently observed in previous studies [13,39,40]. Conversely, the relative abundance of eARGs in water showed a decreasing trend, consistent with the microcosm result. This outcome is based on two key factors according to our model: first, the propensity for iARGs to secrete and release eARGs diminishes with increased nutrient level, as indicated by a smaller *k*1 parameter in our model. This reduction is due to the decreased likelihood of plasmid loss in bacterial hosts carrying ARGs under nutrient-rich conditions, a result corroborated by previous experimental findings [32]. Second, nutrient-rich conditions enhance the potential for horizontal gene transfer, as expressed by the larger *k*2 parameter, which leads to the preferential adsorption of eARGs by bacteria and their subsequent reconversion into iARGs [20].

In the biofilm, however, the trends in the relative abundance of both iARGs and eARGs showed the opposite trend to that in water, consistently with the microcosm result. This is because the biofilm was more nutrient-rich than water and the addition of nutrients would prompt the drug-resistant bacteria to secrete eARGs [20], as expressed by the larger value of the *k*1 parameter in our model. This results in a decreasing trend of iARGs in the biofilm as nutrient levels increase, while the abundance of eARGs increases, matching the trends observed in our microcosm experiments. These results highlight the complex interplay between nutrient levels and ARGs, underscoring the need for further research in dynamic aquatic environments.

### 2.3. Increased Nutrient Delivery Frequencies Boost eARGs’ Abundance and Reduce iARGs

In aquatic environments, nutrient levels are dynamic, often fluctuating due to intensified human activities. For example, wastewater discharge into rivers is a specific mode of nutrient delivery to native microorganisms, with nutrient supply varying from continuous to intermittent based on the discharge duration. To investigate the impact of these variations in nutrient delivery on ARGs, we conducted experiments where the same total amount of nutrients (50 mg COD, 5 mg N, and 1 mg P) was added to microcosms at different dosing frequencies: single administration (1 time/10 d), medium-frequency fractionated administration (5 times/10 d), and high-frequency fractionated administration (20 times/10 d) (Figure 4). The results showed that the relative abundance of iARGs in water was the highest under the single-dosing condition, which was 1.10–1.13 times higher than that of high-frequency nutrient dosing. This can be primarily attributed to the initial nutrient conditions under the single-dosing condition being the highest, which was more favorable for the prompt propagation of antibiotic-resistant bacteria [41,42]. 

By contrast, the eARGs in water showed an increasing trend with the increase of the dosing frequency, and the highest eARG abundance was found under the high-frequency dosing regimen (20 times/10 d), which was 1.35–1.43 times higher than that of lower frequency dosing. This result is due to the fact that the total amount of nutrients remains the same, and small amounts over time allow antibiotic-resistant bacteria to have less nutrients for each period of their growth, which is not conducive to their own reproduction and spread [43], leading to a higher probability of ARGs being released extracellularly.

For ARG on the biofilm, the pattern of change of iARGs was consistent with that of water, and the low-frequency nutrient delivery method was 1.10–2.04 times that of high-frequency nutrient delivery; the pattern of change of eARGs was opposite to that of iARGs, and the high-frequency nutrient delivery (20 times/10 d) was 1.84 times that of the single-delivery situation. Collectively, these results indicate that, in both water and biofilm environments, low-frequency nutrient dosing generally favors the growth and reproduction of iARGs, while high-frequency dosing promotes the relative abundance of eARGs.

To understand the mechanism of how nutrient delivery frequencies affect the propagation of iARGs and eARGs, this parameter was incorporated into our ARGs-nutrient model for simulation (Figure 5). Consistent with the microcosm result, the model showed that the relative abundance of iARGs decreased with increasing frequency of nutrient delivery. This trend is attributed to the fact that lower frequencies of nutrient delivery result in higher nutrient levels during most of the bacterial growth period, as evidenced by higher growth rates in both water and biofilm environments. Such conditions favor the propagation of bacteria hosting ARGs. On the contrary, when nutrients are added multiple times at higher frequencies, the nutrient levels at any given time are lower, resulting in reduced growth rates of bacteria hosting ARGs. This reduction in growth rate leads to a decrease in the relative abundance of iARGs. Additionally, the higher frequency of nutrient delivery reduces the overall nutrient availability, thereby increasing the fitness cost to bacteria hosting ARGs. This is represented by a higher *k*1 parameter in our model, suggesting that, under these conditions, ARGs are more likely to be released into the environment, leading to an increase in the abundance of eARGs.

This study elucidates the complex impact of nutrient variations on ARGs, highlighting the multifaceted interplay among bacterial growth rates, nutrient availability, and processes like horizontal gene transfer and secretion. However, due to time and technical constraints, there are some limitations in this study. First, although we made every effort to simulate realistic environments in the microcosms, they may not fully replicate natural conditions. For example, some small microorganisms may have been excluded due to the filtration method used, and the types of nutrients provided may differ from those in natural environments. Additionally, the establishment of the ARGs-nutrient model, while pioneering, requires further optimization to better reflect the dynamics of microflora in realistic settings. Moreover, the mechanisms by which ARGs respond to nutrient variations need to be explored more deeply using multidimensional approaches. 

Our findings reveal that the propagation trends of iARGs and eARGs are consistently opposite under various conditions, emphasizing the dynamic shifts between ARGs in different physical states. This effect underscores the environmental risks posed by both iARGs and eARGs, with eARGs potentially representing a greater threat due to their ability to survive, spread, and act as gene reservoirs in the environment. Therefore, it is crucial to consider both forms when assessing the overall risk of ARGs. Focusing solely on one type could lead to significant misestimations of the overall ARGs threat. Moreover, the contrasting trends between iARGs and eARGs contribute to a dynamic equilibrium that helps stabilize bacterial resistance levels despite fluctuations in nutrient availability. This may be one reason for the alarming levels of bacterial resistance observed in the environment. The significant levels of bacterial resistance not only pose a threat to human health through the transmission of ARGs but also impact ecosystems by facilitating the spread of ARGs among microbial communities, potentially altering nutrient cycles and even influencing global climate patterns. To mitigate the risk of bacterial resistance, it may be necessary to explore multiple approaches, such as minimizing antibiotic concentrations and other coexisting substances, in addition to adjusting nutrient levels in future studies.

## 3. Materials and Methods 

### 3.1. Sampling and Pretreatment

To prepare microorganisms for the operation of microcosms, water samples were collected from the Nanjing section of the lower Yangtze River (N: 32°02′14.77″, E: 118°70′71.61″). This section, where antibiotics and ARGs have been frequently detected [44,45,46], effectively represents the microflora profile of the Yangtze River under the influence of human activities. To ensure that the microflora was representative of the site and to avoid sampling bias, water was collected from different depths. Specifically, samples were taken from the surface (approximately 0.5 m below the water surface), middle (approximately 4 m below the surface), and bottom (approximately 7.5 m below the surface) using a pump set to the controlled depth of the inlet. These samples from various depths were then thoroughly mixed to create a composite sample with a total volume of 20 L. The water samples were examined for routine physiochemical and microbial qualities as shown in Appendix A. Then, the water sample was filtered through 0.22 µm filter membranes to remove pristine nutrients and antibiotics [47,48]. The filter membranes were washed three times with phosphate buffer solution to rule out the adsorbed nutrients and antibiotics, thereby minimizing potential interference from the original substrate. The membranes were then clipped and transferred into 50 mL centrifuge tubes, filled with PBS, and vertexing occurred for 30 min to facilitate the transfer of microorganisms from filters to the solution. The microorganisms in the solution were then diluted to a total volume of 20 L using sterilized PBS to ensure that the concentration of microorganisms was equivalent to that in the Yangtze River. Subsequently, tetracycline was spiked at a concentration of 1 μg/L, simulating the typical concentration in riverine systems [49,50,51], to maintain the selective pressure within the system.

### 3.2. Construction of Microcosm

A series of one-liter sterile beakers were employed to construct the microcosm system, spiked with 500 mL of microorganism solution, prepared as described above. Three glass slides were placed in each beaker to promote biofilm growth [52]. The microcosm was stirred continuously at room temperature, with speeds ranging from 250 to 300 rpm, to mimic the flow of a natural river. The setting of these conditions and the addition of actual bacteria in the environment will make the changes in the abundance of resistance genes in the microcosm experiment closer to the real environment. To study the effect of nutrient content, three groups were set up by adjusting the dosage of organic carbon (glucose), nitrogen (ammonia nitrogen), and phosphorus (phosphate) to simulate the actual water environment with different nutrient levels: including normal river water (COD = 100 mg/L, COD:N:P = 100:10:2, m/m), polluted water (COD = 333.3 mg/L, COD:N:P = 100:3:0.6, m/m) and eutrophic water (COD = 100 mg/L, COD:N:P = 100:25:5, m/m) [21,22]. To evaluate the effects of nutrient fluctuation [13], as observed during events like wastewater discharge [53], we conducted single administration (1 time/10 d) and split administration (5 times/10 d and 20 times/10 d) in different microcosms under the conditions of a total nutrient (e.g., COD 50 mg, N 5 mg, and P 1 mg). Each microcosm was run for ten days, with all experiments replicated three times.

### 3.3. DNA Extraction

Upon operation, iDNA and eDNA in the planktonic and biofilm on the 1st day and 10th day of microcosms were extracted. For DNA extraction from water, the method has been developed by our group and applied by a few other groups [54,55,56]. Water samples were shaken and filtered through a 0.22 μm pore size filter membrane, with the membrane used for iDNA extraction and the liquid for eDNA extraction. For DNA extraction from biofilm, biofilm on the slides was scraped carefully and transferred into PBS, followed by shaking at 180 rpm for an hour to facilitate the separation between iDNA and eDNA [57]. The suspension was then filtered through a 0.22 μm pore size filter membrane, with the membrane used for iDNA extraction and the liquid for eDNA extraction [19,55]. Although this method does not completely eliminate cross-contamination between iDNA and eDNA, it minimizes it effectively.

For iDNA extraction, the filter was cut into small pieces and placed into a DNA extraction tube and conducted, using the Soil Rapid DNA Extraction Kit (MP Biomedicals, Santa Ana, CA, USA). For eDNA extraction, the method was used accordingly to that previously established [55]. Specifically, the suspension was spiked with 4 mL of Guanidine Hydrochloride Buffer (2 mol/L) and 3 mL of isopropanol, followed by thorough vortexing. Next, 30 μL of suspended magnetic silica beads were introduced into the mixture and gently shaken for 4 min to enable DNA adsorption onto the beads. The beads were then separated using a magnet, and the supernatant was discarded. Subsequently, the beads underwent a washing step with buffer CW (an isopropanol solution containing 7 mol/L guanidine hydrochloride) followed by two washes with 75% ethanol. After washing, the beads were incubated in 30 μL of preheated (55 °C) elution buffer for 10 min, facilitating the transfer of DNA into the solution. Finally, the eDNA was collected by removing the residual beads with a magnet [58]. The extraction efficiency of this method, verified to be between 85.3% and 93.0% [55], is comparable to that of commercial DNA extraction kits.

### 3.4. Quantitative PCR

Based on our previous metagenome analyses of the Yangtze River water samples [59], β-lactam resistance genes, sulfonamide resistance genes, quinolone resistance genes, tetracycline resistance genes, and macrolide resistance genes collectively constituted half of the total abundance of the ARGs, underscores the representativeness of these ARGs in capturing the overall response of ARGs within the system. Quantification of ARGs was performed using quantitative PCR (qPCR) on an ABI 7500 system (Applied Biosystems, Inc. Carlsbad CA, USA). Eleven ARGs (among them *bla_TEM_*, *ermB*, *dfrA*, *qnrS*, *qnrB*, *sul1*, *sul2*, *tetA*, *tetM*, *tetG*, and *tetX*) were selected as relevant indicators of the ARGs’ responses because of their prevalence in aquatic environments, coding for resistance to tetracyclines, β-lactams, sulfonamides, quinolones, and macrolides [10,57]. Primers for ARGs’ quantification have been validated (Appendix A). The qPCR process involved the preparation of ARG standards, calibration, and determination of ARG abundance. First, gene fragments were obtained through PCR and purified by electrophoresis (1% agarose gel) using a DNA purification kit (Omega, Norwalk, CT, USA). The PCR product of each ARG was purified and ligated into the pEASYT1 Simple Cloning (Transgene, Beijing, China) and transformed into competent Escherichia coli DH5α (Transgene, Beijing, China) [60]. Positive clones were selected by blue-white screening and confirmed by PCR and gene sequencing. These procedures were conducted according to the manufacturer’s instructions. The plasmids carrying each ARG were extracted as the standard ARG product. Five consecutive 10-fold dilution gradients were used as templates for qPCR reactions. For the qPCR reactions, a 15 μL reaction system was used, including forward and reverse primers (0.5 μL, 10 μmol of each), 7.5 μL of qPCR Master Mix (TORO Green, Xiamen, Fujian, China), 5.5 μL of ddH_2_O, and 1 μL of template. The following procedure was used: the first phase (94.0 °C, 30 s), the second phase (94.0 °C for 5 s, annealing at set temperature for 15 s (Appendix A), 72 °C for 10 s, 40 cycles), and the third phase (melting process, 95 °C for 15 s, 60 °C for 1 min) [61,62]. The calibration curves were plotted based on the initial concentration of each ARG standard and the number of cycles (CT) of qPCR. The PCR amplification efficiencies were 88–110%, with a detection limit of 1.0 × 10^2^ copies/L. The same process was followed for all samples. Deionized water (ddH_2_O) was used as a negative control, while plasmids containing each ARG were used as positive controls. Each PCR sample was analyzed in triplicate. The absolute abundance of ARGs was calculated based on the cycle times (CT) of the samples, and the relative abundance was calculated based on the ratio of the absolute abundance of ARGs to the 16s rDNA of the same sample [57]. The amplification efficiencies of all qPCRs ranged from 90% to 110%.

### 3.5. ARGs-Nutrient Model Construction

To unravel the propagation of ARGs in response to nutrient variation, referring to previous studies [13], we developed an ARGs-nutrient model based on the Lotka–Volterra model, which is used to describe bacterial growth [63,64], and the Monod equation, which describes the effect of nutrients on growth rate [23]. The model assumes that antibiotic-resistant and sensitive bacteria compete for nutrients, as also suggested previously [65]. Additionally, it considers the potential interconversion between iARGs and eARGs, with iARGs transforming into eARGs through cell lysis or secretion and eARGs reverting to iARGs through uptake by bacteria, based on previous studies [66]. In detail, the growth of bacterial population was assumed to consist of different phenotypes: antibiotic-sensitive strains without plasmids (*y* (1)), and antibiotic-resistant strains with plasmids (*y* (2)), as described in Equations (1) and (2). Logistic growth equations based on the Lotka–Volterra model were used to describe the growth of antibiotic-sensitive and antibiotic-resistant strains [13]. The Monod equation was then included and incorporated into the model to account for the effects of nutrient conditions [24,67,68]. To explore the effect of different nutrients, carbon, and nitrogen functions were separately considered and constructed (Equations (1) and (2)). eDNA primarily comes from the secretion of live bacteria or the lysis of dead bacteria, as described in Equations (3) and (4).
(1)$$dy\left(1\right)=(r1\times \frac{C}{C+M1}\times \frac{N}{N+M2})\times y\left(1\right)\times \left(1-\left(y\left(1\right)+y(2\right)/n\right)+k1\times y\left(2\right)-k2\times y\left(1\right)-a\times y(1)$$
(2)$$dy\left(2\right)=\left(r2\times \frac{C}{C+M1}\times \frac{N}{N+M2}\right)\times y\left(2\right)\times \left(1-\left(y\left(1\right)+y(2\right)/n\right)-k1\times y\left(2\right)+k2\times y\left(1\right)-a\times y(2)$$
(3)$$dy\left(3\right)=k1\times y\left(2\right)+a\times y(2)-b\times y\left(3\right)-k2\times y(1)$$
(4)$$dy\left(4\right)=k1\times y\left(1\right)+a\times y(1)-b\times y\left(4\right)$$
where

*r*1 and *r*2—the maximum growth rates of antibiotic-resistant and antibiotic-sensitive strains, respectively;*k*1 and *k*2—the secretion rate of antibiotic-resistant strains and the transformation frequency of antibiotic-sensitive strains, respectively;*a*—the mortality rate of the strains;*b*—the degradation rate of the plasmids;*C*—the carbon concentration in the nutrients;*N*—the nitrogen concentration in the nutrients;*M*—the half-saturation constant at the maximum growth rate;*n*—the capacity of the strain in the microcosm.

These parameters were selected based on the total number of environmental communities, typical bacterial growth and death rates, and the likelihood of DNA exocytosis, based on previous studies [13,43,66,69,70], as detailed in Appendix A. The model for the propagation of ARGs in biofilm was developed based on the model for planktonic bacteria, as previous studies suggested [43,70], with adjustments to the value of parameters to account for the unique characteristics of biofilms. The operation of the model was conducted using MATLAB.

### 3.6. Data Processing and Statistical Analysis

To assess the variability between the relative abundance of ARGs, we set a significance level of *p* = 0.05. We considered the differential changes in ARGs to be statistically significant only if the *p*-value was less than 0.05. All statistical analysis was conducted by SPSS 15.0.

## 4. Conclusions

We investigated the trends of iARGs and eARGs in response to nutrient variation, using a combination of microcosm experiments and mathematical modeling. Our observations revealed that changes in nutrient levels and dosing frequencies resulted in contrasting responses of iARGs and eARGs in both water and biofilm environments. This opposing behavior underscores the persistence of bacterial resistance in environments with fluctuating nutrients, suggesting that iARGs and eARGs act as reservoirs of bacterial resistance that can impact human health and ecosystems. To effectively mitigate the risk of bacterial resistance, multi-dimensional strategies beyond just nutrient management are necessary.

## Figures and Tables

**Figure 1 antibiotics-13-00817-f001:**
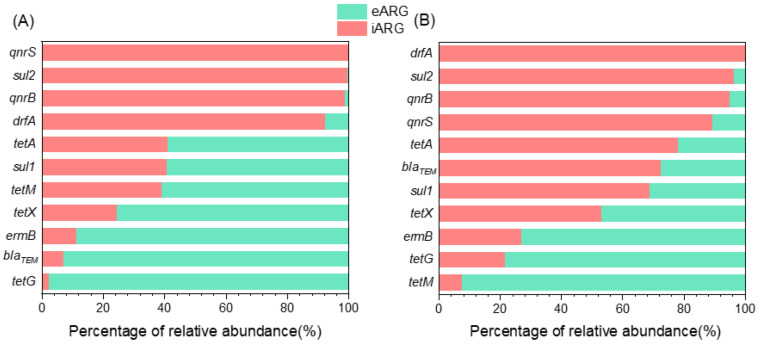
**eARGs are more prevalent in water while iARGs are more prevalent in biofilm.** This figure describes the distribution (relative abundance) of iARGs and eARGs in water (**A**) and biofilm (**B**).

**Figure 2 antibiotics-13-00817-f002:**
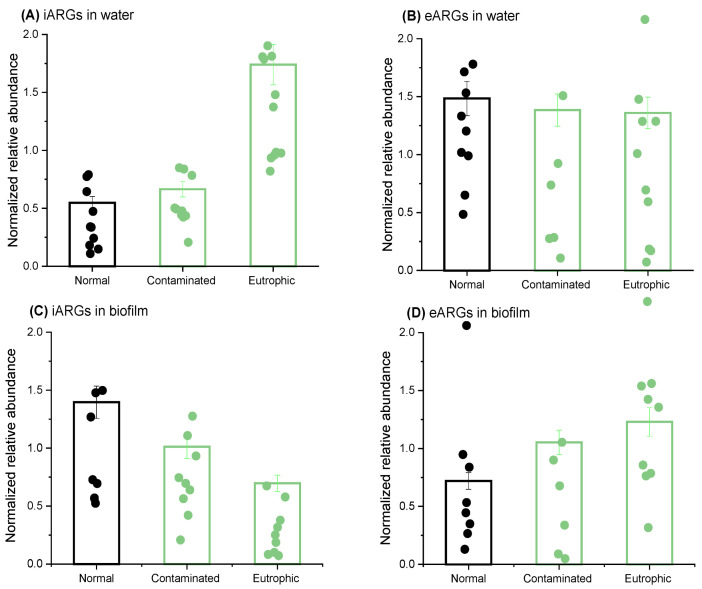
**Differential response of iARGs and eARGs to the variation of nutrient level.** This figure describes changes in the relative abundance of iARGs in water (**A**), eARGs in water (**B**), iARGs in biofilm (**C**), and eARGs in biofilm (**D**) under various nutrient conditions (normal, eutrophic, and contaminated groups) in microcosms. The relative abundance of ARGs in each treatment was normalized to a blank control without nutrient injection. The plots indicate the normalized relative abundance of each single ARG.

**Figure 3 antibiotics-13-00817-f003:**
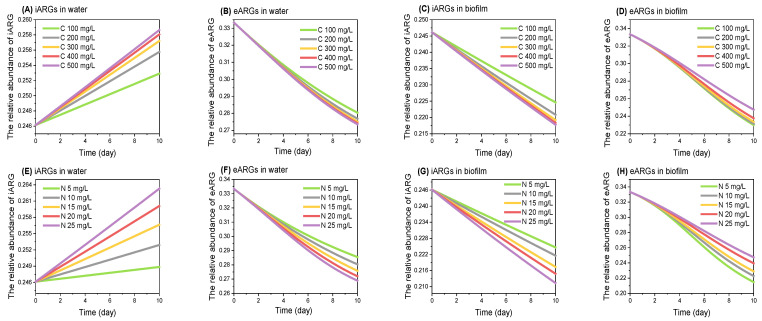
**Simulation of ARG propagation in response to nutrient variations**. This figure illustrates the results from our ARGs-nutrient model, simulating the propagation of ARGs over a 10-day period under varying concentrations of carbon (100–500 mg/L) and nitrogen (5–25 mg/L). The simulations align with our microcosm experiments, demonstrating changes in the relative abundance of iARGs and eARGs. (**A**,**E**) The trends of iARGs in water; (**B**,**F**) the trends of eARGs in water; (**C**,**G**) the trends of iARGs in biofilm; (**D**,**H**) the trends of eARGs in biofilm.

**Figure 4 antibiotics-13-00817-f004:**
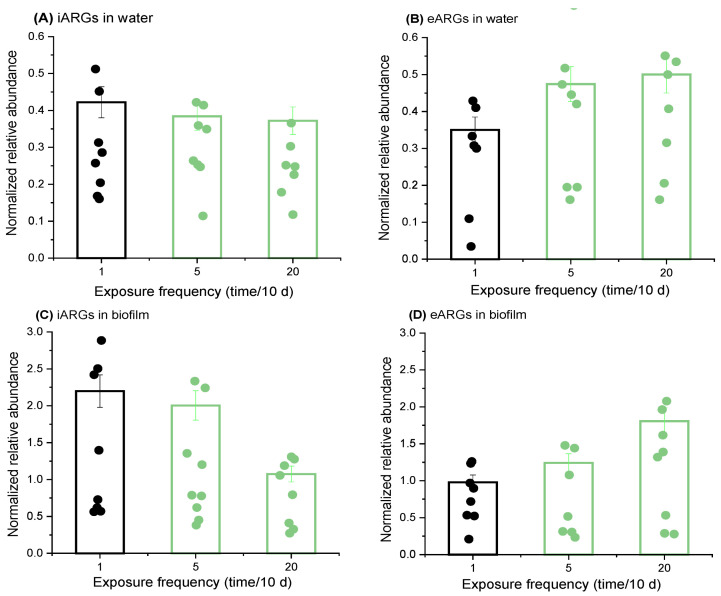
**Differential response of iARGs and eARGs to the frequency of nutrient delivery.** This figure describes changes in the relative abundance of iARGs in water (**A**), eARGs in water (**B**), iARGs in biofilm (**C**), and eARGs in biofilm (**D**) under various frequencies of nutrient delivery. The relative abundance of ARGs in each treatment was normalized to a blank control without nutrient injection. The plots indicate the normalized relative abundance of each single ARG.

**Figure 5 antibiotics-13-00817-f005:**
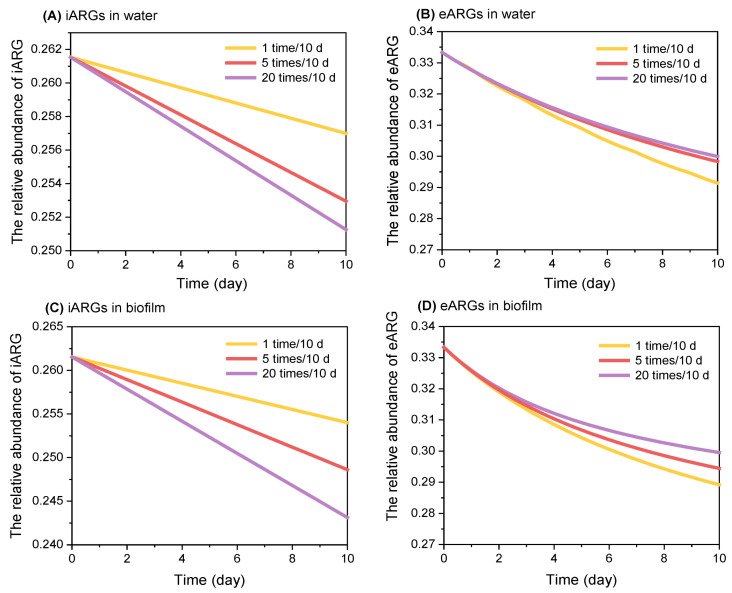
**Simulation of ARG propagation in response to the frequency of nutrient delivery**. This figure illustrates the results from our ARGs-nutrient model, simulating the propagation of ARGs over low medium, and high frequency of nutrient delivery. (**A**) The trends of iARGs in water; (**B**) the trends of eARGs in water; (**C**) the trends of iARGs in biofilm; (**D**) the trends of eARGs in biofilm.

## Data Availability

The original contributions presented in the study are included in the article/Appendix A, and further inquiries can be directed to the corresponding authors.

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
