# Peer review of "Contrasting Dynamics of Intracellular and Extracellular Antibiotic Resistance Genes in Response to Nutrient Variations in Aquatic Environments"

_antibiotics, 2024, doi:10.3390/antibiotics13090817_

Round 1

Reviewer 1 Report

Comments and Suggestions for Authors

In this manuscript, Liu et al. have investigated the changes in the dynamics of intracellular (iARGs) and extracellular (eARGs) with respect to the variations of the nutrient supply in aquatic environments. Overall, this study has some scientific merits, but needs considerable revision to enhance the overall quality.

Comments:

1.       Title: Replace the abbreviation “ARGs” by its full name.

2.       Abstract: Since it is a research-based study, include important quantitative data. At present, the abstract contains mainly the qualitative statements.

3.       Keywords: “Model” is a very generic term, please be specific by mentioning what kind of models were used in this study.

4.       In the first paragraph of the introduction, report the typical concentrations range of the dominant antibiotics found in the aquatic environments (e.g., river).  

5.       In the introduction, provide some combative fundamental knowledge about the “iARGs and eARGs”.

6.       Line 69: “eARGsthat”. Check the typographical error issue here.

7.       In the last paragraph of introduction, describe what are the key hypotheses that were tested in this work.

8.       Line 98: “collected from lower reach of the Yangtze River”. Give the location name (nearby city, province, etc.) where water samples were collected.

9.       Line 99: “collected from the surface, middle-depth, and bottom of the site”. For the middle-depth, any approximate quantitative data on the depth.

10.   Line 101: “move pristine nutrients and antibiotics”. Not easy to understand how to recover dissolved nutrients and antibiotics from water samples.

11.   Line 209: “To assess the variability between the data,”. Elaborate what kinds of data were taken into consideration for statistical analysis.

12.   At the end of results and discussion, include a section describing the important implications and future perspectives of this work.

13.   Statistical analysis: An appropriate statistical analysis should be conducted to show whether the changes in the abundance of iARGs and eARGs under different ecological niche (water vs biofilm) or environmental stress conditions (with or without nutrients, and frequency of nutrients supply) is statistically different.

14.   The reviewer is wondering whether the key physiochemical and microbial qualities of the water samples were measured or not which is an important data for this study, but it looks missing.

15.   There are spacing related errors between the in-text cited reference and the previous word (e.g., resistance(Van Boeckel et al., 2014; Zhang et al., 2015); line 33 – 34). Please fix such issue throughout the manuscript.

Comments on the Quality of English Language

There are several typographical errors exist at present version, thus needs careful and thorough checking in the entire manuscript. 

Author Response

Comments:

  1. Title: Replace the abbreviation “ARGs” by its full name.

Thanks for pointing it out. Has rewritten the title.

“Contrasting Dynamics of Intracellular and Extracellular Antibiotic Resistance Genes in Response to Nutrient Variations in Aquatic Environments” (lines 2~4)

  1. Abstract: Since it is a research-based study, include important quantitative data. At present, the abstract contains mainly the qualitative statements.

Thanks for the comment. We rewrite the abstract and add important quantitative data.

“Our results show that the relative abundance of 7 out of 11 representative eARGs in water exceeds that of iARGs, while 8 iARGs dominate in biofilms. Notably, iARGs and eARGs consistently exhibited opposite, seesaw-like responses to nutrient variation. When nutrient levels increased, iARGs in the water also increased, with the polluted group (COD = 333.3 mg/L, COD: N: P = 100:3:0.6, m/m) and the eutrophic group (COD = 100 mg/L, COD: N: P = 100:25:5, m/m) showing 1.2 and 3.2 times higher levels than the normal group (COD=100 mg/L, COD: N: P = 100:10:2, m/m), respectively. In contrast, eARGs decreased by 6.7% and 8.4% in these groups. On the other hand, in biofilms, higher nutrient levels led to an increase in eARGs by 1.5 and 1.7 times, while iARGs decreased by 17.5% and 50.1% in the polluted and eutrophic groups compared to the normal group. Moreover, while increasing the frequency of nutrient delivery (from 1 time/10 d to 20 times/10 d) generally did not favor iARGs in either water or biofilm, it selectively enhanced eARGs in both.” (lines 16~27)

  1. Keywords: “Model” is a very generic term, please be specific by mentioning what kind of models were used in this study.

Thanks for pointing it out. Has been revised.

“KEYWORDS: Nutrient; Antibiotic resistance genes; Extracellular antibiotic resistance genes; River; Microcosm; ARGs-nutrient model” (lines 35~36)

  1. In the first paragraph of the introduction, report the typical concentrations range of the dominant antibiotics found in the aquatic environments (e.g., river).

Thanks for catching out. More dates have been added.

“In China, the detection levels of various antibiotics in major watersheds are alarmingly high. For instance, macrolide antibiotics have been detected at concentrations ranging from 7.6 to 12,159.4 ng/L, and tetracyclines from 14.1 to 1,676.9 ng/L (Gao et al., 2022; Li et al., 2018; Qiao et al., 2018). (lines 42~45)

  1. In the introduction, provide some combative fundamental knowledge about the “iARGs and eARGs”.

Thanks for catching out. More related content has been added.

“Nutrients impact not only intracellular ARGs (iARGs) within bacterial hosts but also extracellular ARGs (eARGs) that exist outside the cells. iARGs are housed within cells, but when cells undergo lysis or secretion, ARGs are released into the environment, where they can persist for extended periods, especially when adsorbed onto particles (Chowdhury et al., 2021), with their longevity ranging from weeks to years.” (lines 76~80)

  1. Line 69: “eARGsthat”. Check the typographical error issue here.

Thanks for pointing it out. Has been revised.

“The impact of nutrients extends beyond iARGs within bacterial hosts to eARGs that reside outside the cells.”

  1. In the last paragraph of introduction, describe what are the key hypotheses that were tested in this work.

Thanks for catching out. More content has been added.

“Thus, we hypothesize that nutrients will influence the development of bacterial resistance, including both iARGs and eARGs, in distinct ways due to the differing characteristics of these two types of ARGs. To test this hypothesis, we developed a series of microcosms using microflora collected from the Yangtze River, the largest river in Asia.” (lines 90~93)

  1. Line 98: “collected from lower reach of the Yangtze River”. Give the location name (nearby city, province, etc.) where water samples were collected.

Thanks for catching out. The location name has been given.

“To prepare microorganisms for the operation of microcosms, water samples were collected from the Nanjing section of the lower reach of the Yangtze River (N:32°0214.77", E:118°7071.61").” (lines 285~287)

  1. Line 99: “collected from the surface, middle-depth, and bottom of the site”. For the middle-depth, any approximate quantitative data on the depth.

Thanks for catching out. More dates are added.

“To ensure that the microflora was representative of the site and to avoid sampling bias, water was collected from different depths. Specifically, samples were taken from the surface (approximately 0.5 meters below the water surface), middle (approximately 4 meters below the surface), and bottom (approximately 7.5 meters below the surface) using a pump set to the controlled depth of the inlet. These samples from various depths were then thoroughly mixed to create a composite sample with a total volume of 20 liters. The water samples were examined for routine physiochemical and microbial qualities as shown in Table S1.” (lines 289~296)

  1. Line 101: “move pristine nutrients and antibiotics”. Not easy to understand how to recover dissolved nutrients and antibiotics from water samples.

Thanks for the comment. The complex composition of the water samples, including dissolved nutrients and antibiotics, could interfere with the microcosms. To address this, we removed these substances before starting the microcosms by filtering the water through a 0.22 µm filter. This method effectively separates microorganisms from the surrounding media, ensuring that only the desired components (microorganisms for this case) are present in the microcosms. We've rewritten this section to make it clearer.

“The acquired filter membranes were washed with phosphate buffer solution three times to rule out the adsorbed nutrient and antibiotics, thereby minimizing potential interference from the original substrate. The membranes were then clipped and transferred into 50 mL centrifuge tubes, filled with PBS, and vertexing for 30 min to facilitate the transfer of microorganisms from filters to the solution.” (lines 297~302)

  1. Line 209: “To assess the variability between the data,”. Elaborate what kinds of data were taken into consideration for statistical analysis.

Thanks for pointing that out. We have redescribed this section to make it more detailed.

“To assess the variability between the relative abundance of ARGs, we set a significance level of p = 0.05. We considered the differential changes in ARGs to be statistically significant only if the p-value was less than 0.05.” (lines 428~430)

  1. At the end of results and discussion, include a section describing the important implications and future perspectives of this work.

Thanks for pointing it out. Has been added.

“This study elucidates the complex impact of nutrient variations on ARGs, highlighting the multifaceted interplay among bacterial growth rates, nutrient availability, and processes like horizontal gene transfer and secretion.” (lines 256~258)

“Our findings reveal that the propagation trends of iARGs and eARGs are consistently opposite under various conditions, emphasizing the dynamic shifts between ARGs in different physical states. This effect underscores the environmental risks posed by both iARGs and eARGs, with eARGs potentially representing a greater threat due to their ability to survive, spread, and act as gene reservoirs in the environment. Therefore, it is crucial to consider both forms when assessing the overall risk of ARGs. Focusing solely on one type could lead to significant misestimations of the overall ARGs threat. Moreover, the contrasting trends between iARGs and eARGs contribute to a dynamic equilibrium that helps stabilize bacterial resistance levels despite fluctuations in nutrient availability. This may be one reason for the alarming levels of bacterial resistance observed in the environment. The significant levels of bacterial resistance not only pose a threat to human health through the transmission of ARGs but also impact ecosystems by facilitating the spread of ARGs among microbial communities, potentially altering nutrient cycles and even influencing global climate patterns. To mitigate the risk of bacterial resistance, it may be necessary to explore multiple approaches, such as minimizing antibiotic concentrations and other coexisting substances, in addition to adjusting nutrient levels in future studies.” (lines 267~282)

  1. Statistical analysis: An appropriate statistical analysis should be conducted to show whether the changes in the abundance of iARGs and eARGs under different ecological niche (water vs biofilm) or environmental stress conditions (with or without nutrients, and frequency of nutrients supply) is statistically different.

Thanks for pointing it out. Has been revised.

Compared to the normal group, both the polluted and eutrophic groups exhibited higher abundances of iARGs in water, with increases of 1.2 and 3.2 times (p<0.05), respectively.” (lines 141~143)

However, the relative abundance of eARGs in water slightly decreased in the polluted and eutrophic groups, reaching 93.3% and 91.6% of the normal group's levels (p>0.05), respectively.” (lines 144~146)

On the contrary, the relative abundance of iARGs in biofilms generally decreased with increasing nutrient concentration (p<0.05).” (lines 149~151)

Specifically, the mean value of the eleven iARGs abundance for the polluted group decreased by 17.5% compared to the normal group, while the mean value of the eleven iARGs for the eutrophic group decreased by 50.1% (p<0.05).” (lines 151~154)

The relative abundance of eARGs in the polluted and eutrophic groups increased by 1.5-1.7 times compared to the normal group(p<0.05).” (lines 154~156)

The results showed that the relative abundance of iARGs in water tended to increase with elevated carbon (from 100 mg/L to 500 mg/L) and nitrogen levels (from 5 mg/L to 25 mg/L) (p>0.05).” (lines 170~172)

  1. The reviewer is wondering whether the key physiochemical and microbial qualities of the water samples were measured or not which is an important data for this study, but it looks missing.

Thanks for catching out. More dates are added.

“The water samples were examined for routine physiochemical and microbial qualities as shown in Table S1.” (lines 295~296)

Table S1. The routine physiochemical and microbial qualities of the water samples

physiochemical parameters

pH

DO (mg/L)

CODMn (mg/L)

TP (mg/L)

TN (mg/L)

16s rDNA (copies/mL)

values

7.56

6.72

2.7

0.11

1.46

1.43*1010

  1. There are spacing related errors between the in-text cited reference and the previous word (e.g., resistance (Van Boeckel et al., 2014; Zhang et al., 2015); line 33 – 34). Please fix such issue throughout the manuscript.

Thanks for pointing it out. We examined thoroughly and revised.

Reviewer 2 Report

Comments and Suggestions for Authors

Dear Authors,

The manuscript titled "Contrasting Dynamics of Intracellular and Extracellular ARGs in Response to Nutrient Variations in Aquatic Environments" examines how nutrient levels and delivery frequency impact the abundance of intracellular (iARGs) and extracellular antibiotic resistance genes (eARGs). Using microorganisms from the Yangtze River in microcosms, the study found that eARGs dominate in water, while iARGs are more prevalent in biofilms. An increase in nutrients led to a rise in iARGs and a decrease in eARGs in water, with the opposite pattern in biofilms. Frequent nutrient delivery generally increased eARGs but not iARGs. The researchers employed an ARGs-nutrient model, combining Lotka-Volterra and Monod equations, to explore these dynamics. The study highlights the complex interaction between bacterial growth, nutrient availability, and gene transfer, showing a balance between iARGs and eARGs. This dynamic helps maintain bacterial resistance, offering new insights into the persistence of antibiotic resistance in environmental settings.

Here are the main criticisms:

Introduction

1)     The introduction broadly discusses the issue of antibiotic resistance without clearly specifying the unique angle or research question this study addresses. The references to the widespread presence of ARGs in various environments lack specificity regarding which particular aspects are under-explored or novel in this context.

2)     While the section mentions the importance of studying nutrient effects on ARGs, it does not adequately highlight the specific gaps in existing research that this study aims to fill. The statement "the influence of nutrients on this overlooked form of ARGs remains an area of uncertainty" is vague and does not clarify what is unknown or why it is important.

3)     The rationale for choosing specific nutrients, their levels, and dosing frequencies in the study is not sufficiently explained. The justification for the use of microcosm experiments over other methods is not discussed, nor is the relevance of the chosen experimental conditions to real-world scenarios clearly established.

4)     The discussion on the interplay of nutrient levels, bacterial growth, and ARG propagation introduces complex concepts such as "overlap of nutrient ecological niches" and "competitive disadvantage of ARGs-carrying hosts" without sufficient explanation. These terms may be confusing for readers without prior background knowledge.

5)     The introduction does not clearly outline the study's specific objectives or hypotheses. While it mentions various aspects of the study, such as the types of ARGs and the environments considered, it does not specify the expected outcomes or the main questions being investigated.

6)     The significance of understanding the dynamics of iARGs and eARGs in different environments is not thoroughly connected to broader implications for public health or environmental management. The potential practical applications or policy implications of the research findings are not adequately discussed.

Materials and Methods

1)    The description of the sampling process is somewhat vague, particularly in explaining why samples were collected from specific depths and how these samples are representative of the broader environment. Additionally, the rationale for using the particular site on the Yangtze River is not explained.

2)    The methods used for removing nutrients and antibiotics from water samples lack detail on potential impacts on the microbial community. There is also no mention of how these steps might influence the results, especially in relation to naturally occurring ARGs.

3)    The setup and conditions of the microcosms are described without providing sufficient justification for the chosen nutrient concentrations and ratios. It is unclear why specific concentrations of glucose, ammonia nitrogen, and phosphate were selected to represent "normal," "polluted," and "eutrophic" water conditions.

4)    The methodology does not explain why specific administration frequencies were chosen for nutrient dosing, nor how these frequencies mimic real-world conditions.

5)    While the process for extracting iDNA and eDNA is described, there is limited discussion on the potential for cross-contamination between these two forms or how the procedures ensure their distinct separation.

6)    The section lacks a critical evaluation of the DNA extraction efficiency and potential biases introduced by the methods used.

7)    The selection of ARGs for quantification is mentioned without explaining the criteria for their selection or how they are representative of the broader ARG pool. The primer specificity and validation are also not discussed, which is critical for ensuring accurate and reliable qPCR results. The description of the qPCR setup, including the reaction conditions and controls, is not detailed enough, potentially affecting reproducibility.

8)    The description of the ARGs-nutrient model lacks clarity on the assumptions made in the model and how well these assumptions match real-world conditions. There is no discussion on the potential limitations of the model or how variations in key parameters were accounted for.

9)    The choice of using the Lotka-Volterra model and Monod equation is not justified in the context of ARG propagation, and there is no discussion on why these models were considered appropriate for the study.

Results and Discussion

1)     The discussion frequently makes broad statements without sufficient specificity or quantitative detail. For example, the phrase "high prevalence of eARGs" is used without providing specific comparative figures or benchmarks.

2)     The section often refers to increases or decreases in ARGs without consistently providing statistical significance or confidence intervals. The lack of explicit mention of statistical tests used to validate these findings weakens the credibility of the conclusions drawn.

3)     While the section references previous studies, it does not sufficiently compare or contrast the study's findings with existing literature. There is a need for a deeper discussion on how the observed trends align or diverge from established knowledge and what new insights are provided.

4)     The explanation of mechanisms, such as the "propensity for iARGs to secrete and release eARGs" or the effects of nutrient levels on horizontal gene transfer, lacks depth. The discussion does not sufficiently explore the underlying biological or chemical processes that may be driving these observations.

5)     The use of the ARGs-nutrient model and the parameters (such as k1 and k2) are presented without adequate justification or validation. There is little discussion on the assumptions made in the model, how the parameters were derived, or their real-world applicability.

6)     Some conclusions, such as the "seesaw effect" and its implications for environmental ARG risk assessment, seem speculative and are not robustly supported by the data. The claim that focusing on one type of ARG (iARGs or eARGs) could lead to misestimations lacks clear evidence and a detailed argument.

7)     The discussion repeatedly emphasizes the "novel perspective" offered by the study without clearly delineating what is novel or how it advances the field. This emphasis on novelty sometimes overshadows the need for rigorous validation and cautious interpretation of results.

8)     The section does not sufficiently discuss the practical implications of the findings for managing antibiotic resistance in aquatic environments. It also lacks a clear outline of potential future research directions that could build on the current study's findings.

Conclusions

1)    The conclusions restate the findings from the results section without delving into the broader implications or significance of these findings. There is a lack of specificity in terms of how these observations contribute to the existing body of knowledge or how they could inform practical applications.

2)    The section emphasizes the observed "seesaw-like response" of iARGs and eARGs without sufficiently interpreting the underlying mechanisms or potential reasons for these patterns. There is no in-depth discussion on why these opposite trends occur or what they suggest about the ecology of ARGs in different environments.

3)    The statement about the "high potential ecological and human health risk under the One Health perspective" is made without sufficient qualification or evidence. The conclusions seem to generalize the risks associated with ARGs without acknowledging the limitations of the study or the need for further research to confirm these risks.

4)    The section does not address the limitations of the study, such as the constraints of using microcosm experiments and mathematical modeling, which may not fully capture real-world complexities. There is also no discussion of potential confounding factors or uncertainties in the data.

5)    There is no mention of how these findings could be applied in environmental management or policy-making. Additionally, the section lacks suggestions for future research directions that could further explore or validate the study's findings.

6)    The conclusions largely reiterate what has already been covered in the results and discussion, without offering new insights or a compelling final takeaway. The emphasis on the prevalence of eARGs and iARGs does not offer a clear concluding statement or highlight a novel perspective.

Comments on the Quality of English Language

Moderate editing of English language required

Author Response

The manuscript titled "Contrasting Dynamics of Intracellular and Extracellular ARGs in Response to Nutrient Variations in Aquatic Environments" examines how nutrient levels and delivery frequency impact the abundance of intracellular (iARGs) and extracellular antibiotic resistance genes (eARGs). Using microorganisms from the Yangtze River in microcosms, the study found that eARGs dominate in water, while iARGs are more prevalent in biofilms. An increase in nutrients led to a rise in iARGs and a decrease in eARGs in water, with the opposite pattern in biofilms. Frequent nutrient delivery generally increased eARGs but not iARGs. The researchers employed an ARGs-nutrient model, combining Lotka-Volterra and Monod equations, to explore these dynamics. The study highlights the complex interaction between bacterial growth, nutrient availability, and gene transfer, showing a balance between iARGs and eARGs. This dynamic helps maintain bacterial resistance, offering new insights into the persistence of antibiotic resistance in environmental settings.

We sincerely thank the reviewer for the valuable suggestions on our manuscript. All comments have been carefully considered, and the manuscript has been revised accordingly.

Here are the main criticisms:

Introduction

  • The introduction broadly discusses the issue of antibiotic resistance without clearly specifying the unique angle or research question this study addresses. The references to the widespread presence of ARGs in various environments lack specificity regarding which particular aspects are under-explored or novel in this context.

Thank you for the insightful comment. The primary novelty of our study lies in understanding the fate of ARGs, particularly extracellular ARGs, which have been previously underestimated, within microflora in aquatic environments under varying nutrient conditions—one of the most crucial factors influencing ARGs. While a few studies have touched on this topic, most have been conducted under pure culture lab conditions or through theoretical analysis, which do not fully reflect the dynamics of microflora in realistic environments. We have clarified and emphasized these points in the revised manuscript.

“However, these conclusions are predominantly based on pure cultured laboratory conditions or through theoretical modeling analysis. The impact of nutrient concentrations and exposure frequencies on natural aquatic microflora remains poorly understood, highlighting the need for microcosm studies or field research that accurately replicates actual environmental conditions.” (lines 70~75)

“Nutrients impact not only intracellular ARGs (iARGs) within bacterial hosts but also extracellular ARGs (eARGs) that exist outside the cells. iARGs are housed within cells, but when cells undergo lysis or secretion, ARGs are released into the environment, where they can persist for extended periods, especially when adsorbed onto particles (Chowdhury et al., 2021), with their longevity ranging from weeks to years.” (lines 76~80)

“Thus, we hypothesize that nutrients will influence the development of bacterial resistance, including both iARGs and eARGs, in distinct ways due to the differing characteristics of these two types of ARGs. To test this hypothesis, we developed a series of microcosms using microflora collected from the Yangtze River, the largest river in Asia.” (lines 90~93)

  • While the section mentions the importance of studying nutrient effects on ARGs, it does not adequately highlight the specific gaps in existing research that this study aims to fill. The statement "the influence of nutrients on this overlooked form of ARGs remains an area of uncertainty" is vague and does not clarify what is unknown or why it is important.

Thank you for the comment. As mentioned earlier, we have reorganized the relevant text to better highlight the importance and novelty of our study. Specifically, we have clarified the specific gaps in existing research that this study aims to fill, particularly regarding the influence of nutrients on extracellular ARGs, an area that remains underexplored and uncertain. We have now detailed what is unknown and why addressing these uncertainties is crucial for advancing our understanding of ARG dynamics in aquatic environments.

“However, these conclusions are predominantly based on pure cultured laboratory conditions or through theoretical modeling analysis. The impact of nutrient concentrations and exposure frequencies on natural aquatic microflora remains poorly understood, highlighting the need for microcosm studies or field research that accurately replicates actual environmental conditions.” (lines 70~75)

“Nutrients impact not only intracellular ARGs (iARGs) within bacterial hosts but also extracellular ARGs (eARGs) that exist outside the cells. iARGs are housed within cells, but when cells undergo lysis or secretion, ARGs are released into the environment, where they can persist for extended periods, especially when adsorbed onto particles (Chowdhury et al., 2021), with their longevity ranging from weeks to years.” (lines 76~80)

“Thus, we hypothesize that nutrients will influence the development of bacterial resistance, including both iARGs and eARGs, in distinct ways due to the differing characteristics of these two types of ARGs. To test this hypothesis, we developed a series of microcosms using microflora collected from the Yangtze River, the largest river in Asia.” (lines 90~93)

  • The rationale for choosing specific nutrients, their levels, and dosing frequencies in the study is not sufficiently explained. The justification for the use of microcosm experiments over other methods is not discussed, nor is the relevance of the chosen experimental conditions to real-world scenarios clearly established.

Thanks for catching out. Has been added.

“We then explored the colonization and development of intracellular and extracellular ARGs in planktonic and biofilm forms under varying levels of organic matter, nitrogen, and phosphorus nutrients, including conditions representative of normal river water(COD=100 mg/L, COD: N: P = 100:10:2, m/m), polluted water(COD = 333.3 mg/L, COD: N: P = 100:3:0.6, m/m), and eutrophic water(COD = 100 mg/L, COD: N: P = 100:25:5, m/m), reflecting the water quality found in rivers under different conditions (Yu et al., 2017; Zhao et al., 2019). Additionally, to account for the dynamic nature of nutrient fluctuations in real environments, such as the intermittent discharge of wastewater into rivers, we simulate nutrient dosing frequencies, including single administration (1 time/10 d) and split administration (5 times/10 d and 20 times/10 d), while maintaining consistent total nutrient content.(lines 93~103)

“The microcosm was stirred continuously at room temperature, with speeds ranging from 250 to 300 rpm, to mimic the flow of a natural river. The setting of these conditions and the addition of actual bacteria in the environment will make the changes in the abundance of resistance genes in the microcosm experiment closer to the real environment.” (lines 310~314)

“To study the effect of nutrient content, three groups were set up by adjusting the dosage of organic carbon (glucose), nitrogen (ammonia nitrogen), and phosphorus (phosphate) to simulate the actual water environment with different nutrient levels (Yu et al., 2017; Zhao et al., 2019): including normal river water (COD=100 mg/L, COD: N: P = 100:10:2, m/m), polluted water (COD = 333.3 mg/L, COD: N: P = 100:3:0.6, m/m) and eutrophic water (COD = 100 mg/L, COD: N: P = 100:25:5, m/m). To evaluate the effects of nutrient fluctuation (Letten et al., 2021), as observed during events like wastewater discharge (Taylor et al., 2020), we conducted single administration (1 time/10 d) and split administration (5 times/10 d and 20 times/10 d) in different microcosms under the conditions of a total nutrient (e.g., COD 50 mg, N 5 mg, and P 1 mg).” (lines 314~323)

“In this study, we established three scenarios: a normal group representing typical water conditions in rivers, a polluted group reflecting conditions frequently reported in urban inland rivers, and a eutrophic group characterizing nutrient-rich environments (Figure 2) (Danyang et al., 2021; Janicka et al., 2022).” (lines 138~141)

  • The discussion on the interplay of nutrient levels, bacterial growth, and ARG propagation introduces complex concepts such as "overlap of nutrient ecological niches" and "competitive disadvantage of ARGs-carrying hosts" without sufficient explanation. These terms may be confusing for readers without prior background knowledge.

Thanks for pointing out. Have reorganized the relevant text to make it more explicit.

“For example, it has been reported that the adaptation cost of host-carrying ARGs increases as nutrient availability decreases (Letten et al., 2021). This is because hosts carrying ARGs and their non-resistant counterparts often compete for the same nutrients, leading to heightened interspecific competition as nutrient availability declines (Pereira and Berry, 2017; Silva et al., 2012). (lines 58~62)

“Furthermore, the fluctuation of nutrient levels, rather than their mere abundance, plays a significant role in shaping antibiotic resistance patterns by increasing the adaptive cost of antibiotic-resistant bacteria and decreasing their abundance. This leads to the disadvantage during competition (Letten et al., 2021).” (lines 67~70)

  • The introduction does not clearly outline the study's specific objectives or hypotheses. While it mentions various aspects of the study, such as the types of ARGs and the environments considered, it does not specify the expected outcomes or the main questions being investigated.

Thank you for the comment. In the revised manuscript, we have clearly outlined the key hypotheses and emphasized their importance, along with the study's specific objectives. This addition ensures that the expected outcomes and main research questions are clearly articulated.

“Thus, we hypothesize that nutrients will influence the development of bacterial resistance, including both iARGs and eARGs, in distinct ways due to the differing characteristics of these two types of ARGs. To test this hypothesis, we developed a series of microcosms using microflora collected from the Yangtze River, the largest river in Asia.” (lines 90~93)

“The translational relationship between the development of ARGs and nutrient conditions provides valuable insights into the persistence of bacterial resistance in real-world environments. These findings also offer important implications for reducing the potential risk of ARG spread to humans and ecosystems.” (lines 106~109)

  • The significance of understanding the dynamics of iARGs and eARGs in different environments is not thoroughly connected to broader implications for public health or environmental management. The potential practical applications or policy implications of the research findings are not adequately discussed.

Thanks for the comment. We have addressed the importance and implications in the revised manuscript.

“The translational relationship between the development of ARGs and nutrient conditions provides valuable insights into the persistence of bacterial resistance in real-world environments. These findings also offer important implications for reducing the potential risk of ARG spread to humans and ecosystems.” (lines 106~109)

“This study elucidates the complex impact of nutrient variations on ARGs, highlighting the multifaceted interplay among bacterial growth rates, nutrient availability, and processes like horizontal gene transfer and secretion. However, due to time and technical constraints, there are some limitations in this study. First, although we made every effort to simulate realistic environments in the microcosms, they may not fully replicate natural conditions. For example, some small microorganisms may have been excluded due to the filtration method used, and the types of nutrients provided may differ from those in natural environments. Additionally, the establishment of the ARG-nutrient model, while pioneering, requires further optimization to better reflect the dynamics of microflora in realistic settings. Moreover, the mechanisms by which ARGs respond to nutrient variations need to be explored more deeply using multidimensional approaches. “(lines 256~266)

Materials and Methods

  • The description of the sampling process is somewhat vague, particularly in explaining why samples were collected from specific depths and how these samples are representative of the broader environment. Additionally, the rationale for using the particular site on the Yangtze River is not explained.

Thanks for the comment. This information has been clarified in the revised manuscript.

“To prepare microorganisms for the operation of microcosms, water samples were collected from the Nanjing section of the lower Yangtze River (N:32°02′14.77", E:118°70′71.61"). This section, where antibiotics and ARGs have been frequently detected (Hu et al., 2023; Rajasekar et al., 2023; Zhang et al., 2020), effectively represents the microflora profile of the Yangtze River under the influence of human activities. (lines 285~289)

“To ensure that the microflora was representative of the site and to avoid sampling bias, water was collected from different depths. Specifically, samples were taken from the surface (approximately 0.5 meters below the water surface), middle (approximately 4 meters below the surface), and bottom (approximately 7.5 meters below the surface) using a pump set to the controlled depth of the inlet. These samples from various depths were then thoroughly mixed to create a composite sample with a total volume of 20 liters. The water samples were examined for routine physiochemical and microbial qualities as shown in Table S1. (lines 289~296)

  • The methods used for removing nutrients and antibiotics from water samples lack detail on potential impacts on the microbial community. There is also no mention of how these steps might influence the results, especially in relation to naturally occurring ARGs.

Thanks for the comment. We used membrane filtration to separate the microbial community from the pristine substrate, a method commonly adopted in previous studies (Mao et al., 2014; Yuan et al., 2019). While it is true that some small microorganisms may be missed by this method, resulting in a slightly different microflora composition compared to that in the water, we believe this is acceptable for our study as the influence is consistent across all microcosms with different nutrient regimes. We have discussed the potential impact of this in the revised manuscript.

“First, although we made every effort to simulate realistic environments in the microcosms, they may not fully replicate natural conditions. For example, some small microorganisms may have been excluded due to the filtration method used, and the types of nutrients provided may differ from those in natural environments.” (lines 259~263)

  • The setup and conditions of the microcosms are described without providing sufficient justification for the chosen nutrient concentrations and ratios. It is unclear why specific concentrations of glucose, ammonia nitrogen, and phosphate were selected to represent "normal," "polluted," and "eutrophic" water conditions.

Thanks for pointing that out. We have clarified the rationale behind the chosen nutrient concentrations and ratios in the revised manuscript, providing supporting literature to justify the selection of glucose, ammonia nitrogen, and phosphate levels used to represent "normal," "polluted," and "eutrophic" water conditions.

“We then explored the colonization and development of intracellular and extracellular ARGs in planktonic and biofilm forms under varying levels of organic matter, nitrogen, and phosphorus nutrients, including conditions representative of normal river water(COD=100 mg/L, COD: N: P = 100:10:2, m/m), polluted water(COD = 333.3 mg/L, COD: N: P = 100:3:0.6, m/m), and eutrophic water(COD = 100 mg/L, COD: N: P = 100:25:5, m/m), reflecting the water quality found in rivers under different conditions (Yu et al., 2017; Zhao et al., 2019) (lines 93~99)

  • The methodology does not explain why specific administration frequencies were chosen for nutrient dosing, nor how these frequencies mimic real-world conditions.

Thanks for pointing that out. Actually, we explained in the result and discussion part. In the revised manuscript we also clarified in the method section to address the relationality.

“Additionally, to account for the dynamic nature of nutrient fluctuations in real environments, such as the intermittent discharge of wastewater into rivers, we simulate nutrient dosing frequencies, including single administration (1 time/10 d) and split administration (5 times/10 d and 20 times/10 d), while maintaining consistent total nutrient content.” (lines 99~103)

“In aquatic environments, nutrient levels are dynamic, often fluctuating due to intensified human activities. For example, wastewater discharge into rivers is a specific mode of nutrient delivery to native microorganisms, with nutrient supply varying from continuous to intermittent based on the discharge duration.” (lines 201~204)

  • While the process for extracting iDNA and eDNA is described, there is limited discussion on the potential for cross-contamination between these two forms or how the procedures ensure their distinct separation.

Thanks for the comment. We recognize that there is a possibility of cross-contamination, though a series of approach was applied to minimize it. During the isolation process, 0.22 um filtration was used to collect cells as much as possible. Also, multiple washes are performed for membranes to remove possible attached eDNA on cells.

“Although this method does not completely eliminate cross-contamination between iDNA and eDNA, it minimizes it effectively.” (lines 334~336)

  • The section lacks a critical evaluation of the DNA extraction efficiency and potential biases introduced by the methods used.

Thanks for catching out. The extraction efficiency of DNA has been added to the text.

“The extraction efficiency of this method, verified to be between 85.3% and 93.0%(Yuan et al., 2019) , is comparable to that of commercial DNA extraction kits. (lines 349~351)

  • The selection of ARGs for quantification is mentioned without explaining the criteria for their selection or how they are representative of the broader ARG pool. The primer specificity and validation are also not discussed, which is critical for ensuring accurate and reliable qPCR results. The description of the qPCR setup, including the reaction conditions and controls, is not detailed enough, potentially affecting reproducibility.

Thanks for pointing it out. We added accordingly.

“Based on our previous metagenome analyses of the Yangtze River water samples (Liu et al., 2023), β-lactam resistance genes, sulfonamide resistance genes, quinolone resistance genes, tetracycline resistance genes, and macrolide resistance genes collectively constituted half of the total abundance of the ARGs, underscores the representativeness of these ARGs in capturing the overall response of ARGs within the system.” (lines 353~357)

“Primers for ARGs quantification have been validated (Table S2).” (lines 362~363)

“The PCR product of each ARG was purified and ligated into the pEASYT1 Simple Cloning (Transgene, Beijing, China) and transformed into competent Escherichia coli DH5α (Transgene, Beijing, China) (Zeng et al., 2020). Positive clones were selected by blue-white screening and confirmed by PCR and gene sequencing. These procedures were conducted according to the manufacturer's instructions. The plasmids carrying each ARG were extracted as the standard ARG product.” (lines 366~371)

“Deionized water (ddH2O) was used as a negative control, while plasmids containing each ARG were used as positive controls.” (lines 380~382)

“The amplification efficiencies of all qPCRs ranged from 90% to 110%.” (lines 385~386)

  • The description of the ARGs-nutrient model lacks clarity on the assumptions made in the model and how well these assumptions match real-world conditions. There is no discussion on the potential limitations of the model or how variations in key parameters were accounted for.

Thank you for the insightful comment. We have revised the manuscript to clarify the assumptions made in the ARGs-nutrient model. Specifically, the model assumes that antibiotic-resistant and sensitive bacteria compete for the same nutrient resources and that iARGs and eARGs can interconvert through processes like cell lysis and secretion. We acknowledge that these assumptions while simplifying real-world complexities, are necessary to develop a tractable model. We have also expanded the discussion to address the potential limitations of the model, such as the challenges in fully capturing the diversity of microbial communities and environmental conditions.

“The model assumes that antibiotic-resistant and sensitive bacteria compete for nutrients, as also suggested previously (Wang, 2020). Additionally, it considers the potential interconversion between iARGs and eARGs, with iARGs transforming into eARGs through cell lysis or secretion and eARGs reverting to iARGs through uptake by bacteria, based on previous studies (Zhang et al., 2022).. (lines 391~395)

“Additionally, the establishment of the ARGs-nutrient model, while pioneering, requires further optimization to better reflect the dynamics of microflora in realistic settings.” (lines 263~265)

  • The choice of using the Lotka-Volterra model and Monod equation is not justified in the context of ARG propagation, and there is no discussion on why these models were considered appropriate for the study.

Thank you for pointing this out. We have reorganized this section to make it more accurate.

“To unravel the propagation of ARGs in response to nutrient variation, referring to previous studies (Letten et al., 2021), we developed an ARGs-nutrient model based on the Lotka-Volterra model, which is used to describe bacterial growth (Lax et al., 2020; Maslov and Sneppen, 2017), and the Monod equation, which describes the effect of nutrients on growth rate (Mao and Lu, 2016).” (lines 388~391)

Results and Discussion

  • The discussion frequently makes broad statements without sufficient specificity or quantitative detail. For example, the phrase "high prevalence of eARGs" is used without providing specific comparative figures or benchmarks.

Thank you for pointing this out.  We have revised the discussion to enhance specificity.

“The results reveal that eARGs are more prevalent in water compared to iARGs, with seven out of eleven ARGs being present in this form.” (lines 114~115)

  • The section often refers to increases or decreases in ARGs without consistently providing statistical significance or confidence intervals. The lack of explicit mention of statistical tests used to validate these findings weakens the credibility of the conclusions drawn.

Thanks for pointing it out. Has been revised.

Compared to the normal group, both the polluted and eutrophic groups exhibited higher abundances of iARGs in water, with increases of 1.2 and 3.2 times (p<0.05), respectively.” (lines 141~143)

However, the relative abundance of eARGs in water slightly decreased in the polluted and eutrophic groups, reaching 93.3% and 91.6% of the normal group's levels (p>0.05), respectively.” (lines 144~146)

On the contrary, the relative abundance of iARGs in biofilms generally decreased with increasing nutrient concentration (p<0.05).” (lines 149~151)

Specifically, the mean value of the eleven iARGs abundance for the polluted group decreased by 17.5% compared to the normal group, while the mean value of the eleven iARGs for the eutrophic group decreased by 50.1% (p<0.05).” (lines 151~154)

The relative abundance of eARGs in the polluted and eutrophic groups increased by 1.5-1.7 times compared to the normal group(p<0.05).” (lines 154~156)

The results showed that the relative abundance of iARGs in water tended to increase with elevated carbon (from 100 mg/L to 500 mg/L) and nitrogen levels (from 5 mg/L to 25 mg/L) (p>0.05).” (lines 170~172)

  • While the section references previous studies, it does not sufficiently compare or contrast the study's findings with existing literature. There is a need for a deeper discussion on how the observed trends align or diverge from established knowledge and what new insights are provided.

Thanks for pointing that out. Previous literature has been added and compared.

“This observation indicates that eARGs are widespread and prevalent in aquatic environments and their relative abundance is higher than that of iARGs, which is consistent with previously conducted studies (Yuan et al., 2023; Zhang et al., 2018). (lines 116~119)

“This trend, aligning with the microcosm experiment, is likely due to nutrient promoting the proliferation of drug-resistant bacteria while reducing their adaptive fitness costs, a phenomenon consistently observed in previous studies (Letten et al., 2018; Letten et al., 2021; Ryu et al., 2018).” (lines 172~175)

  • The explanation of mechanisms, such as the "propensity for iARGs to secrete and release eARGs" or the effects of nutrient levels on horizontal gene transfer, lacks depth. The discussion does not sufficiently explore the underlying biological or chemical processes that may be driving these observations.

Thanks for the comment.  We have added possible explanations for these mechanisms in the revised text. However, as our study primarily uses a model to reflect the underlying processes, we acknowledge that this approach may be somewhat superficial. We have also addressed these limitations in the revised manuscript to provide a more balanced discussion.

“This reduction is due to the decreased likelihood of plasmid loss in bacterial hosts carrying ARGs under nutrient-rich conditions, a result corroborated by previous experimental findings (Zarei-Baygi and Smith, 2021).” (lines 179~181)

“Moreover, the mechanisms by which ARGs respond to nutrient variations need to be explored more deeply using multidimensional approaches.” (lines 265~266)

  • The use of the ARGs-nutrient model and the parameters (such as k1 and k2) are presented without adequate justification or validation. There is little discussion on the assumptions made in the model, how the parameters were derived, or their real-world applicability.

Thanks for the insightful comment. We have added necessary references and discussions in the revised manuscript.

“To unravel the propagation of ARGs in response to nutrient variation, referring to previous studies (Letten et al., 2021), we developed an ARGs-nutrient model based on the Lotka-Volterra model, which is used to describe bacterial growth (Lax et al., 2020; Maslov and Sneppen, 2017), and the Monod equation, which describes the effect of nutrients on growth rate (Mao and Lu, 2016). (lines 388~391)

“These parameters were selected based on the total number of environmental communities, typical bacterial growth and death rates, and the likelihood of DNA exocytosis, based on previous studies (Brander et al., 2023; Jiang et al., 2022; Jin et al., 2020; Letten et al., 2021; Zhang et al., 2022), as detailed in Table S3.

(lines 420~422)

  • Some conclusions, such as the "seesaw effect" and its implications for environmental ARG risk assessment, seem speculative and are not robustly supported by the data. The claim that focusing on one type of ARG (iARGs or eARGs) could lead to misestimations lacks clear evidence and a detailed argument.

Thanks for the comment. The "seesaw effect" and its implications for environmental ARG risk assessment are drawn from the results of this study. We acknowledge that some conclusions may appear speculative, so we have revised the text to more explicitly connect the observed data with our interpretations.

“This contrasting response between iARGs and eARGs contributes to a dynamic balance that stabilizes bacterial resistance levels amid nutrient fluctuations.” (lines 31~33)

“Our observations revealed that changes in nutrient levels and dosing frequencies resulted in contrasting responses of iARGs and eARGs in both water and biofilm environments.” (lines 435~437)

  • The discussion repeatedly emphasizes the "novel perspective" offered by the study without clearly delineating what is novel or how it advances the field. This emphasis on novelty sometimes overshadows the need for rigorous validation and cautious interpretation of results.

Thank you for your feedback. We have revised the text to clearly articulate what is novel about our study and how it advances the field. In doing so, we have also balanced the emphasis on novelty with a more rigorous validation and cautious interpretation of our results, making the discussion more convincing.

“This study offers helpful implications on the persistence of bacterial resistance in the environment.” (lines 33~34)

“Moreover, the contrasting trends between iARGs and eARGs contribute to a dynamic equilibrium that helps stabilize bacterial resistance levels despite fluctuations in nutrient availability. This may be one reason for the alarming levels of bacterial resistance observed in the environment. The significant levels of bacterial resistance not only pose a threat to human health through the transmission of ARGs but also impact ecosystems by facilitating the spread of ARGs among microbial communities, potentially altering nutrient cycles and even influencing global climate patterns.” (lines 273~280)

  • The section does not sufficiently discuss the practical implications of the findings for managing antibiotic resistance in aquatic environments. It also lacks a clear outline of potential future research directions that could build on the current study's findings.

Thanks for pointing that out. Revisions have been made.

“To mitigate the risk of bacterial resistance, it may be necessary to explore multiple approaches, such as minimizing antibiotic concentrations and other coexisting substances, in addition to adjusting nutrient levels in future studies.” (lines 280~282)

Conclusions

  • The conclusions restate the findings from the results section without delving into the broader implications or significance of these findings. There is a lack of specificity in terms of how these observations contribute to the existing body of knowledge or how they could inform practical applications.

Thanks for pointing that out. We have reorganized the section.

“We investigated the trends of iARGs and eARGs in response to nutrient variation, using a combination of microcosm experiments and mathematical modeling. Our observations revealed that changes in nutrient levels and dosing frequencies resulted in contrasting responses of iARGs and eARGs in both water and biofilm environments. This opposing behavior underscores the persistence of bacterial resistance in environments with fluctuating nutrients, suggesting that iARGs and eARGs act as reservoirs of bacterial resistance that can impact human health and ecosystems. To effectively mitigate the risk of bacterial resistance, multi-dimensional strategies beyond just nutrient management are necessary.” (lines 434~442)

  • The section emphasizes the observed "seesaw-like response" of iARGs and eARGs without sufficiently interpreting the underlying mechanisms or potential reasons for these patterns. There is no in-depth discussion on why these opposite trends occur or what they suggest about the ecology of ARGs in different environments.

Thanks for pointing that out.  We've refined the expression and expanded the discussion in the revised text. We've also included potential reasons for these opposite trends, exploring what they might indicate about the ecology of ARGs in various environments, while acknowledging the limitations of our current understanding.

“Our observations revealed that changes in nutrient levels and dosing frequencies resulted in contrasting responses of iARGs and eARGs in both water and biofilm environments. This opposing behavior underscores the persistence of bacterial resistance in environments with fluctuating nutrients, suggesting that iARGs and eARGs act as reservoirs of bacterial resistance that can impact human health and ecosystems. To effectively mitigate the risk of bacterial resistance, multi-dimensional strategies beyond just nutrient management are necessary.” (lines 435~442)

  • The statement about the "high potential ecological and human health risk under the One Health perspective" is made without sufficient qualification or evidence. The conclusions seem to generalize the risks associated with ARGs without acknowledging the limitations of the study or the need for further research to confirm these risks.

Thanks for pointing that out. We've refined the expression and acknowledge the limitations of our current understanding in the revised text.

“However, due to time and technical constraints, there are some limitations in this study. First, although we made every effort to simulate realistic environments in the microcosms, they may not fully replicate natural conditions. For example, some small microorganisms may have been excluded due to the filtration method used, and the types of nutrients provided may differ from those in natural environments. Additionally, the establishment of the ARGs-nutrient model, while pioneering, requires further optimization to better reflect the dynamics of microflora in realistic settings. Moreover, the mechanisms by which ARGs respond to nutrient variations need to be explored more deeply using multidimensional approaches. “(lines 258~266)

  • The section does not address the limitations of the study, such as the constraints of using microcosm experiments and mathematical modeling, which may not fully capture real-world complexities. There is also no discussion of potential confounding factors or uncertainties in the data.

Thanks for pointing that out. We've added a discussion on the limitations in the revised text

“This study elucidates the complex impact of nutrient variations on ARGs, highlighting the multifaceted interplay among bacterial growth rates, nutrient availability, and processes like horizontal gene transfer and secretion. However, due to time and technical constraints, there are some limitations in this study. First, although we made every effort to simulate realistic environments in the microcosms, they may not fully replicate natural conditions. For example, some small microorganisms may have been excluded due to the filtration method used, and the types of nutrients provided may differ from those in natural environments. Additionally, the establishment of the ARG-nutrient model, while pioneering, requires further optimization to better reflect the dynamics of microflora in realistic settings. Moreover, the mechanisms by which ARGs respond to nutrient variations need to be explored more deeply using multidimensional approaches. (lines 256~266)

  • There is no mention of how these findings could be applied in environmental management or policy-making. Additionally, the section lacks suggestions for future research directions that could further explore or validate the study's findings.

Tahnks for the comment. We have refined the text accordingly.

“This study elucidates the complex impact of nutrient variations on ARGs, highlighting the multifaceted interplay among bacterial growth rates, nutrient availability, and processes like horizontal gene transfer and secretion. However, due to time and technical constraints, there are some limitations in this study. First, although we made every effort to simulate realistic environments in the microcosms, they may not fully replicate natural conditions. For example, some small microorganisms may have been excluded due to the filtration method used, and the types of nutrients provided may differ from those in natural environments. Additionally, the establishment of the ARG-nutrient model, while pioneering, requires further optimization to better reflect the dynamics of microflora in realistic settings. Moreover, the mechanisms by which ARGs respond to nutrient variations need to be explored more deeply using multidimensional approaches. (lines 256~266)

“To mitigate the risk of bacterial resistance, it may be necessary to explore multiple approaches, such as minimizing antibiotic concentrations and other coexisting substances, in addition to adjusting nutrient levels in future studies. (lines 280~282)

“We investigated the trends of iARGs and eARGs in response to nutrient variation, using a combination of microcosm experiments and mathematical modeling. Our observations revealed that changes in nutrient levels and dosing frequencies resulted in contrasting responses of iARGs and eARGs in both water and biofilm environments. This opposing behavior underscores the persistence of bacterial resistance in environments with fluctuating nutrients, suggesting that iARGs and eARGs act as reservoirs of bacterial resistance that can impact human health and ecosystems. To effectively mitigate the risk of bacterial resistance, multi-dimensional strategies beyond just nutrient management are necessary.” (lines 434~442)

  • The conclusions largely reiterate what has already been covered in the results and discussion, without offering new insights or a compelling final takeaway. The emphasis on the prevalence of eARGs and iARGs does not offer a clear concluding statement or highlight a novel perspective.

Thanks for the insightful comment. We have reorganized the text accordingly.

“We investigated the trends of iARGs and eARGs in response to nutrient variation, using a combination of microcosm experiments and mathematical modeling. Our observations revealed that changes in nutrient levels and dosing frequencies resulted in contrasting responses of iARGs and eARGs in both water and biofilm environments. This opposing behavior underscores the persistence of bacterial resistance in environments with fluctuating nutrients, suggesting that iARGs and eARGs act as reservoirs of bacterial resistance that can impact human health and ecosystems. To effectively mitigate the risk of bacterial resistance, multi-dimensional strategies beyond just nutrient management are necessary.” (lines 434~442)

Round 2

Reviewer 1 Report

Comments and Suggestions for Authors

The submitted revision to the comments looks good. The quality of the of study is improved after revision. 

Comments on the Quality of English Language

The English language has been improved compared to previous version. 

Reviewer 2 Report

Comments and Suggestions for Authors

Dear Authors,

The manuscript “Contrasting Dynamics of Intracellular and Extracellular Antibiotic Resistance Genes in Response to Nutrient Variations in Aquatic Environments” investigates how nutrient levels and delivery frequency impact the propagation of antibiotic resistance genes (ARGs) in aquatic environments, using microorganisms from the Yangtze River. The findings reveal that intracellular ARGs (iARGs) and extracellular ARGs (eARGs) exhibit opposite responses to nutrient variations: increased nutrients raise iARGs in water but decrease them in biofilms, while the opposite occurs for eARGs. The research developed a model integrating bacterial growth dynamics to explain this seesaw-like behavior, highlighting the complex interactions that stabilize bacterial resistance in fluctuating environments.

According to my opinion, the manuscript can be published in its current form.

Comments on the Quality of English Language

Minor editing of English language required.